# Environmental Mechanisms Influencing the Pathogenesis and Progression of Type 1 Diabetes

**DOI:** 10.3390/ijms262311613

**Published:** 2025-11-30

**Authors:** Yuntao Tang, Weizhou Wang, Zhengsha Huang, Chenxi Zhang, Jia Zhang, Yafang Pang, Shangze Li

**Affiliations:** 1School of Medicine, Chongqing University, Chongqing 400030, China; 13397221357@163.com (Y.T.); weizhouwang27@163.com (W.W.); 202437021011@stu.cqu.edu.cn (C.Z.); zhangjia011223@163.com (J.Z.); pyf187842144362023@126.com (Y.P.); 2College of Medicine, Tibet University, Lhasa 850000, China; 18886207417@163.com

**Keywords:** Type 1 diabetes, environmental factors, pathogenesis, gene-environment interaction, immune dysregulation, oxidative stress, epigenetic modification, gut microbiota, viral infection, pancreatic β-cell

## Abstract

Type 1 diabetes (T1D) is a complex autoimmune disorder characterized by the selective destruction of insulin-producing pancreatic β-cells. While genetic predisposition establishes susceptibility, environmental factors play a pivotal role in triggering and modulating the autoimmune process. This narrative review synthesizes current evidence from epidemiological, clinical, and experimental studies to elucidate the mechanisms by which environmental exposures influence the pathogenesis and progression of T1D. We discuss how persistent organic pollutants, heavy metals, air pollutants, viral infections, and gut microbiome alterations contribute to β-cell dysfunction, loss of immune tolerance, and enhanced autoimmunity. Our analysis reveals that these environmental triggers act through multiple interconnected pathways, including oxidative stress, mitochondrial dysfunction, epigenetic modifications, innate immune activation, and gut barrier disruption. Understanding these mechanisms provides critical insights for developing preventive strategies and targeted interventions to mitigate environmental risks. These findings underscore the importance of addressing environmental exposures as modifiable risk factors, offering a potential framework for early interventions aimed at preventing or slowing T1D progression in at-risk populations.

## 1. Introduction

Type 1 diabetes (T1D) is an autoimmune disorder where the immune system selectively destroys insulin-producing pancreatic β-cells, resulting in insulin deficiency and the need for lifelong insulin therapy. Genetic factors, especially variations in the human leukocyte antigen (*HLA*) complex, such as *HLA-DR3-DQ2* and *HLA-DR4-DQ8*, contribute significantly to disease risk. However, the incomplete concordance in monozygotic twins (30–50%) suggests that environmental factors also play a crucial role in disease onset and progression [1,2,3].

Over the past decades, rapid industrialization, global environmental changes, and lifestyle transitions have intensified exposure to a diverse array of environmental stressors. Among these, environmental pollutants and viral infections have emerged as the two most influential categories shaping susceptibility to Type 1 Diabetes (T1D). Persistent organic pollutants (POPs)—such as bisphenol A (BPA), polychlorinated biphenyls (PCBs), and organochlorine pesticides—act alongside airborne pollutants (e.g., PM_2.5_, ozone), toxic heavy metals (e.g., lead, cadmium), and essential trace elements (e.g., iron, zinc, copper). These stressors disrupt β-cell function and immune homeostasis through mechanisms including oxidative stress, mitochondrial dysfunction, and epigenetic reprogramming. Similarly, viral infections, particularly enteroviruses (e.g., Coxsackievirus B) and, more recently, Severe Acute Respiratory Syndrome Coronavirus 2(SARS-CoV-2,) can directly damage β-cells, induce molecular mimicry, or perturb the gut–pancreas axis, collectively driving the autoimmune process (Figure 1).

Given the heterogeneity of findings across epidemiological, clinical, and experimental research, a systematic synthesis of evidence is essential to clarify how these exposures converge mechanistically to trigger β-cell autoimmunity. We identified relevant articles mainly from PubMed, Embase, Web of Science, and the Cochrane Library, focusing on publications from approximately 2012 to 2025. Representative high-quality epidemiological studies, mechanistic experiments, and recent systematic reviews were prioritized to illustrate key concepts.

This review integrates mechanistic and epidemiological insights to delineate how environmental pollutants and viral infections interact with host genetics and epigenetics to promote β-cell destruction and immune dysregulation. By consolidating evidence across molecular, cellular, and systemic levels, we aim to illuminate key pathogenic pathways—including oxidative stress, mitochondrial dysfunction, epigenetic remodeling, and gut barrier disruption—and to inform the design of preventive strategies and targeted interventions for mitigating the environmental burden of T1D worldwide.

## 2. Environmental Pollutants and T1D Susceptibility

### 2.1. Persistent Organic Pollutants (POPs)

Persistent organic pollutants (POPs), including bisphenol A (BPA), polychlorinated biphenyls (PCBs), and organochlorine pesticides, are associated with T1D through mechanisms such as oxidative stress, mitochondrial dysfunction, and immune dysregulation. These pollutants disturb metabolic and immune balance, leading to β-cell dysfunction and promoting autoimmunity. Systematic reviews have confirmed a significant link between T1D and certain POPs, emphasizing their role in disease development [4].

#### 2.1.1. Experimental Studies

Animal and in vitro studies have confirmed the causal role of POPs in T1D development, demonstrating dose- and model-specific effects.

Bisphenol A (BPA), a ubiquitous persistent organic pollutant (POP) used in polycarbonate plastics and epoxy resins, contributes to type 1 diabetes (T1D) pathogenesis by disrupting metabolic and immune homeostasis. As an endocrine-disrupting chemical (EDC), BPA interferes with hormonal signaling pathways essential for pancreatic β-cell function and immune regulation, thereby creating a microenvironment conducive to autoimmune destruction of insulin-producing β-cells [4,5]. In experimental models, BPA exhibits model-specific and dose-dependent actions. In multiple low-dose streptozotocin (MLDSTZ)–induced C57BL/6 mice, BPA administered in drinking water at 0.1 mg/L increased T1D incidence in a dose-dependent manner: lower concentrations reduced CD4^+^ T cells and elevated pro-inflammatory cytokines, whereas higher concentrations predominantly altered cytokine profiles [6,7]. In non-obese diabetic (NOD) mice, BPA is typically delivered orally at 30 or 300 µg/kg/day over exposure periods ranging from 16 days to more than 150 days. Under these conditions, BPA accelerates T1D onset in females by inducing pro-inflammatory gut microbiome (GMB) shifts and delays disease onset in males via reductions in both pro- and anti-inflammatory GMB components, with immunomodulation identified as the primary mechanism [8]. Moreover, exposure of NOD mice to a POP mixture containing BPA shifts the serum metabolome toward increased sugar derivatives and triacylglycerols, alongside reduced long-chain fatty acids and membrane-associated lipid classes (e.g., phosphatidylcholines and sphingomyelins), mirroring metabolic signatures previously associated with T1D susceptibility in humans (Table 1) [9].

#### 2.1.2. Human Studies

Polychlorinated biphenyls (PCBs) and organochlorine pesticides (e.g., p,p′-DDE, trans-nonachlor) are strongly associated with elevated T1D risk, particularly in pediatric populations. Notably, the SEARCH for Diabetes in Youth Case Control Study (SEARCH-CC) investigated this link in 442 youths (112 controls, 182 with T1D and normal insulin sensitivity [T1D/IS], and 148 with T1D and insulin resistance [T1D/IR]). The study revealed that higher plasma concentrations of p,p′-DDE (2nd tertile: OR 2.0, 95% CI 1.0–3.8; 3rd tertile: OR 2.4, 95% CI 1.2–5.0), trans-nonachlor (2nd tertile: OR 2.5, 95% CI 1.3–5.0; 3rd tertile: OR 2.3, 95% CI 1.1–5.1), and PCB-153 (3rd tertile: OR 2.3, 95% CI 1.1–4.6) were significantly associated with T1D/IS, but not T1D/IR, compared to controls.

In vitro experiments using INS-1E pancreatic β-cells further validated the direct toxic effects of these POPs: treatment with PCB-153 or p,p′-DDE at environmentally relevant concentrations (1 × 10^−15^ M to 5 × 10^−6^ M) impaired β-cell ability to produce and secrete insulin in response to glucose, which was accompanied by reduced mRNA expression of genes critical for insulin synthesis (*Ins1*, *Ins2*), glucose sensing (*Slc2a2*, *Gck*), and insulin secretion (*Abcc8*, *Kcnj11*) (Table 1) [10].

#### 2.1.3. Mechanistic Insights

The mechanisms underlying POP-induced T1D are multifaceted, primarily involving direct pancreatic β-cell dysfunction, immune dysregulation, and oxidative stress. First, POPs activate the aryl hydrocarbon receptor (AhR) pathway—a ligand-dependent transcription factor regulating cytochrome P450 enzymes. This activation drives oxidative stress and systemic inflammation, creating an environment conducive to β-cell damage and autoimmune activation [5,9]. Second, POPs exhibit time-dependent cytotoxicity: while acute exposure (e.g., 2 days) to PCB-153 or p,p′-DDE has minimal impact on survival, chronic exposure progressively induces β-cell death, thereby depleting the functional insulin-secreting mass [10]. Third, POPs disrupt systemic metabolic homeostasis beyond the pancreas. By interfering with adipogenesis and receptor signaling, they indirectly exacerbate metabolic stress on β-cells, accelerating T1D progression [5]. Additionally, POPs may alter epigenetic patterns (e.g., DNA methylation, histone modifications) in genes related to immune regulation and β-cell function, though further studies are needed to clarify the extent of this contribution (Figure 2 and Table 1) [5,9].

### 2.2. Metal and Trace Elements

Type 1 diabetes mellitus (T1DM) is an autoimmune disease resulting from the complex interplay of genetic susceptibility and environmental triggers. Beyond traditional factors, metal elements and trace elements influence critical processes including immune regulation, insulin secretion and signaling, and oxidative stress, thereby affecting T1D pathogenesis and progression.

#### 2.2.1. Iron

Iron imbalance significantly impacts T1D risk and progression. A systematic review suggested that intrauterine iron exposure during pregnancy is associated with an increased risk of developing islet autoimmunity (IA) in offspring [11,12]. The mechanism likely involves iron-induced oxidative stress, which triggers β-cell apoptosis and increases the immunogenicity of islet antigens. The TEDDY study found a U-shaped relationship between dietary iron intake and IA risk [13,14]. In children with a genetic predisposition to iron overload, high iron intake significantly raised IA risk. This gene–environment interaction suggests that iron may exacerbate autoimmune processes in genetically predisposed individuals by promoting free radical generation and inflammation. Beyond disease onset, iron status influences complications. Mendelian randomization studies indicate a causal relationship where elevated ferritin levels increase the risk of T1D with renal complications, possibly by promoting renal fibrosis and inflammation via oxidative pathways [15]. Conversely, higher total iron-binding capacity (TIBC), often indicative of lower iron stores, is associated with a reduced risk. Furthermore, vitamin D deficiency may exacerbate iron deficiency anemia in T1D children by upregulating the iron-regulatory hormone hepcidin, which blocks iron absorption and mobilization, creating a vicious cycle of inflammation and anemia (Table 1) [16].

#### 2.2.2. Zinc

Zinc is vital for immune system integrity and insulin metabolism. Zinc deficiency is strongly linked to T1D development [17,18]. Mechanistically, zinc acts as a key modulator of innate and adaptive immunity; it is essential for the function of regulatory T-cells (Tregs), which suppress autoimmunity, and its deficiency can skew the immune system towards a pro-inflammatory state. Furthermore, zinc is crucial for insulin crystallization, storage, and secretion within pancreatic β-cells. In therapeutics, zinc’s properties are harnessed in innovative delivery systems. A “smart” glucose-responsive microneedle patch utilizes zinc to form a complex with glucagon (Z-GCN). The mechanism of release involves competitive binding: at low glucose levels, boronic acid groups in the patch dissociate from glucose and preferentially bind to catechol groups, displacing and releasing Z-GCN to counteract hypoglycemia [19,20].

#### 2.2.3. Copper

Copper metabolism is frequently dysregulated in T1D. Elevated serum copper levels are observed in newly diagnosed patients. As a cofactor for various enzymes, copper imbalance can disrupt metabolic homeostasis. While copper is involved in antioxidant defense (e.g., via superoxide dismutase), excess copper can paradoxically promote oxidative stress through Fenton-like reactions. Review articles note that abnormal copper levels contribute to insulin resistance [18], potentially by interfering with insulin signaling pathways. Animal studies demonstrate that chronic exposure to high-dose copper chloride exacerbates vascular dysfunction in diabetes by increasing oxidative stress, reducing nitric oxide bioavailability, and inducing inflammation in perivascular adipose tissue, thereby accelerating vascular complications (Table 1) [21].

#### 2.2.4. Toxic Heavy Metals

Exposure to toxic heavy metals like lead and cadmium is a significant environmental risk factor. A study on Egyptian children with T1D found a correlation between elevated hair lead levels and poorer glycemic control (higherHbA1c) [22]. Lead is known to impair mitochondrial function and insulin signaling, while also promoting oxidative stress. Cadmium exposure, often linked to parental smoking, can accumulate in tissues and damage pancreatic β-cells (Table 1).

#### 2.2.5. Other Trace Elements

Other elements play diverse roles. Selenium levels are often elevated in new-onset T1D [15]. As a component of glutathione peroxidase, selenium is critical for antioxidant defense, but its precise role in T1D pathogenesis remains unclear and may be complex. Vanadium compounds exhibit insulin-mimetic properties in animal models. Concomitant administration with insulin improved endothelial dysfunction in diabetic rats [23,24], likely by inhibiting protein tyrosine phosphatases and enhancing insulin receptor signaling pathways. Gold nanoparticles (GNPs) represent a technological application. When loaded with a proinsulin peptide and injected intradermally, GNPs recruit and expand antigen-specific T-cells in the skin [25]. This mechanism offers a platform for studying immune responses and developing antigen-specific immunotherapies.

### 2.3. Air Pollutants and Particulate Matter

Air pollutants are increasingly recognized as important environmental modifiers of T1D risk and disease course. Epidemiological and experimental data indicate that inhaled pollutants influence T1D pathogenesis through disruption of immune homeostasis, metabolic dysregulation, and direct or indirect injury to pancreatic islets. Below, we summarize the pollutant-specific effects of particulate matter (PM_2.5_/PM_10_), ozone (O_3_), and Asian sand dust (ASD) on T1D susceptibility and progression, integrating evidence from gestational exposures, childhood cohorts, and animal models.

#### 2.3.1. Particulate Matter (PM_2.5_/PM_10_)

Particulate matter (PM_2.5_ and PM_10_) exerts multifaceted effects on T1D, spanning fetal programming, metabolic control in established disease, and islet toxicity. Long-term maternal exposure to ambient air pollution during pregnancy has been associated with a higher risk of islet autoimmunity and T1D in offspring. Refs. [26,27] These associations suggest that PM may perturb fetal immune development and tolerance acquisition, thereby lowering the threshold for β-cell–directed autoimmunity later in life.

In children and adolescents with established T1D, chronic exposure to elevated residential levels of particulate matter (PM_2.5_/PM_10_) is linked to a deterioration in metabolic control. Evidence from a large longitudinal cohort of 44,383 pediatric patients indicates that those residing in the highest PM_2.5_/PM_10_ quartiles exhibited modest but statistically significant increases in HbA1c, higher rates of severe hypoglycemia, and slightly lower daily insulin requirements compared to those in the lowest quartiles [28]. Crucially, these associations persisted even after adjusting for potential confounders—including sex, age, age at onset, treatment modality, migration background, urbanization degree, and socioeconomic deprivation—thereby supporting an independent association between PM exposure and compromised metabolic control. However, these population-based analyses warrant cautious interpretation due to inherent limitations. Exposure assessment relied on average concentrations at the 5-digit postcode level rather than individual-level monitoring. Consequently, residual confounding from co-exposed pollutants or personal lifestyle factors could not be fully excluded, potentially leading to exposure misclassification or an overestimation of association strength. Nevertheless, these findings suggest that PM exposure aggravates glycemic variability and may clinically alter insulin sensitivity or requirements in T1D patients.

Experimental studies provide mechanistic insight into how PM_2.5_ interacts with diabetes-related susceptibility. In C57BL/6J mice, pre-existing T1D or diet-induced obesity sensitizes animals to PM_2.5_-induced lung injury, characterized by neutrophilic inflammation, alveolar wall thickening, enhanced lipid peroxidation, and macrophage infiltration. Transcriptomic and metabolomic analyses indicate that PM_2.5_ activates nuclear receptor (NR)-regulated inflammatory and senescence pathways, while simultaneously inhibiting glutathione (GSH)-mediated detoxification [29]. These systemic pro-oxidative and pro-inflammatory signals can propagate beyond the lung and contribute to β-cell stress and damage, consistent with the inherently low antioxidant capacity of β-cells.

In addition to β-cells, particulate matter–induced oxidative stress may impair islet α-cell function. Oxidative and inflammatory stressors have been shown to perturb TMEM55A-mediated phosphatidylinositol 5-phosphate (PI5P) signaling, a pathway crucial for actin cytoskeleton remodeling and glucagon secretion by α-cells [30]. Disruption of this axis may exacerbate dysregulated glucagon release in T1D, thereby worsening glycemic instability and counter-regulatory responses.

Epidemiologically, higher ambient PM_10_ levels have also been associated with increased incidence of pediatric T1D. In a regional study from Apulia (Italy), children living in areas within the highest PM_10_ tertile exhibited higher T1D incidence compared with those in the lowest tertile, supporting a link between chronic PM exposure, impaired immune tolerance, and β-cell damage at the population level [31]. Collectively, these observations indicate that PM_2.5_/PM_10_ influence T1D through a combination of developmental immune programming, systemic inflammation and oxidative stress, altered insulin sensitivity, and direct or indirect disruption of islet cell function (Table 1).

#### 2.3.2. Ozone (O_3_)

Ozone (O_3_), a major photochemical air pollutant, also appears to modulate T1D risk, particularly when exposure occurs during critical developmental windows. Maternal exposure to higher ambient O_3_ during gestation has been associated with a borderline increase in T1D risk among offspring, likely reflecting interference with fetal immune maturation and thymic education. Altered differentiation and functional programming of immune cells in utero may predispose children to loss of tolerance to β-cell antigens.

Beyond direct effects on immune development, prenatal O_3_ exposure has been implicated in the development of gestational diabetes mellitus (GDM). A meta-analysis reported that higher O_3_ levels—especially during the first trimester—are associated with a modest but significant increase in GDM risk [30]. Maternal GDM, in turn, alters the intrauterine metabolic milieu, including glucose, lipid, and inflammatory profiles. Such changes may epigenetically program offspring toward enhanced metabolic vulnerability, systemic low-grade inflammation, and altered β-cell resilience, thereby indirectly increasing T1D susceptibility in genetically at-risk children [30].

Mechanistically, O_3_ is a potent oxidant that induces ROS formation and airway inflammation, which can spill over into the systemic circulation. Chronic oxidative stress and inflammatory mediators derived from O_3_-exposed lung tissue may act on peripheral immune organs and pancreatic islets, complementing PM-mediated pathways in driving β-cell stress and autoimmunity. Thus, O_3_ and PM frequently coexist as components of complex air pollution mixtures, and their combined effects on T1D risk likely reflect convergent oxidative and immunomodulatory mechanisms (Table 1).

#### 2.3.3. Asian Sand Dust (ASD)

Asian sand dust (ASD) represents a distinct type of airborne particulate exposure, composed primarily of mineral particles originating from desert regions, often coated with environmental pollutants and microorganisms. Interestingly, experimental data in NOD mice suggest that ASD may exert context-dependent immunomodulatory effects that differ from those of urban PM. Intratracheal administration of ASD in NOD mice delayed cyclophosphamide (CY)-induced T1D onset. Ref. [32] Adoptive transfer experiments showed that splenocytes from ASD-treated donors conferred partial protection against diabetes in recipient NOD mice, indicating a systemic reprogramming of immune responses.

Mechanistically, ASD exposure increased IFN-γ production while reducing the proportion of regulatory T cells (Tregs) [32]. Although IFN-γ is typically considered a pro-inflammatory cytokine, in this specific model, the net effect favored a protective immune phenotype against β-cell-directed autoimmunity, possibly by altering the balance and activation thresholds of diabetogenic effector T cells and antigen-presenting cells. These findings highlight that not all inhaled particulates uniformly promote T1D; rather, their impact depends on particle composition, dose, timing, and the underlying immune context (Table 1).

Taken together, studies on PM_2.5_/PM_10_, O_3_, and ASD underscore that air pollutants influence T1D through multiple, partially overlapping mechanisms, including developmental immune programming, systemic oxidative stress and inflammation, perturbation of islet β- and α-cell function, and complex immunomodulatory effects [26,27,28,29,30,31,32].

This table summarizes the proposed mechanisms linking select environmental pollutants to T1D pathogenesis, which are often derived from in vitro or animal model studies and may contribute to human disease through interconnected pathways.

## 3. Viral Infections: Key Triggers of β-Cell Autoimmunity

Viral infections are recognized environmental triggers of T1D, causing disease through direct β-cell damage, immune dysregulation, and interactions with the gut-pancreas axis. Enteroviruses, particularly Coxsackievirus B (CVB), are most commonly associated with T1D, while recent studies also suggest that SARS-CoV-2 and other viruses play a role in disease initiation and progression [33,34,35,36].

### 3.1. Direct β-Cell Damage and Dysfunction

Viruses can directly infect pancreatic β-cells, leading to cytolysis, functional impairment, and release of autoantigens that initiate autoimmune responses. CVB can infect human islets, with single-cell RNA sequencing studies revealing infection of both endocrine and exocrine cells—though ductal cells exhibit the strongest interferon (IFN) response, suggesting indirect β-cell damage from neighboring infected cells [37,38]. Direct viral infection triggers intrinsic immune responses within β-cells, including the production of type I IFNs. While antiviral, this IFN response induces endoplasmic reticulum (ER) stress and mitochondrial dysfunction, further compromising β-cell viability and insulin secretion [39,40]. For example, IFN-α stimulation leads to heterogeneous production of mitochondrial reactive oxygen species (mtROS) in human β-cells, a marker of cellular stress that enhances immunogenicity (Figure 3) [41].

### 3.2. Immune-Mediated Mechanisms

Viruses disrupt immune tolerance through molecular mimicry and bystander activation. Molecular mimicry occurs when viral peptides share structural homology with pancreatic β-cell antigens: CVB encodes a capsid protein (VP1) with sequence similarity to glutamic acid decarboxylase 65 (*GAD65*), triggering cross-reactive autoreactive T-cells that attack β-cells [42,43,44]. Bystander activation involves viral-induced pro-inflammatory environments (e.g., in pancreatic draining lymph nodes) that lower the activation threshold of dormant autoreactive T-cells. Viral infections upregulate chemokines such as CXCL10, recruiting Th1 lymphocytes to islets and perpetuating autoimmune inflammation [44].

Notably, the immune outcome of viral infections is context-dependent. In some mouse models, viral exposure activates a protective axis involving invariant natural killer T (iNKT) cells and plasmacytoid dendritic cells (pDCs), leading to Treg generation and suppression of diabetogenic effector cells (Figure 3) [45]. This highlights the dual role of viruses as both triggers and potential protectors, depending on the host’s immune status and genetic background.

### 3.3. Gut-Virus-Pancreas Crosstalk

Viral infections disrupt the gut microbiome (dysbiosis), impair gut barrier integrity, and promote bacterial translocation to pancreatic lymph nodes—creating a pro-autoimmune environment. CVB4 infection in mice restructures the gut microbiome (reducing *Bifidobacteria* and *Akkermansia*) prior to T1D onset, and transfer of this “diabetogenic” microbiome to naive mice enhances T1D susceptibility [46,47,48,49]. Additionally, viral-induced gut barrier dysfunction allows microbial metabolites and endotoxins (e.g., lipopolysaccharide) to enter the systemic circulation, triggering chronic low-grade inflammation that primes the immune system for β-cell attack (Figure 3) [50,51].

### 3.4. Emerging Viral Triggers

The COVID-19 pandemic revealed a significant increase in new-onset pediatric T1D and diabetic ketoacidosis (DKA), with a systematic review reporting a 9.5% rise in pediatric T1D incidence during the first year of the pandemic [52]. The mechanisms underlying the association between SARS-CoV-2 infection and T1D appear to be multifactorial. First, SARS-CoV-2 can directly damage pancreatic β-cells by binding to receptors such as (Angiotensin-Converting Enzyme 2 )ACE2 expressed on β-cells, allowing viral entry and replication. This leads to structural and functional alterations, including reduced insulin secretory granules, impaired proinsulin/insulin processing, β-cell transdifferentiation or dedifferentiation, and even direct β-cell loss. Second, the local and systemic inflammatory responses induced by SARS-CoV-2 result in the release of pro-inflammatory cytokines that trigger β-cell apoptosis, activate bystander T cells, and disrupt peripheral immune tolerance, thereby initiating autoimmunity. This is particularly relevant because individuals with T1D already exhibit chronic low-grade inflammation (elevated IL-1α, IL-1β, IL-2, IL-6, TNF-α), which may exacerbate this process. Third, oxidative stress serves as a central mediator: extreme glucose excursions, a hallmark of T1D, induce oxidative stress through pathways such as protein kinase C activation and increased formation of advanced glycation end products (AGEs), while SARS-CoV-2 infection further amplifies oxidative damage. Additional mechanisms, including viral persistence, molecular mimicry, and activation of endogenous human retroviruses, may also contribute to SARS-CoV-2-associated T1D [53,54,55,56,57,58,59]. Other viruses, such as human endogenous retroviruses, have also been implicated in T1D pathogenesis, potentially via immune activation and molecular mimicry [60].

## 4. Gut Microenvironment in T1D Regulation

The gut microenvironment—comprising the microbiota, intestinal epithelial barrier, and microbial metabolites—serves as a critical orchestrator of immune homeostasis. Dysregulation of this ecosystem is increasingly recognized as a pivotal driver in T1D pathogenesis, precipitating the loss of tolerance to pancreatic β-cells through interconnected pathways [61,62].

### 4.1. Gut Microbiota Dysbiosis and Metabolite Alterations

A hallmark of T1D is gut microbiota dysbiosis, characterized by reduced microbial diversity and profound compositional shifts. This imbalance typically features a depletion of beneficial short-chain fatty acid (SCFA)-producing commensals (e.g., *Faecalibacterium prausnitzii*, *Roseburia intestinalis*) alongside an enrichment of pro-inflammatory taxa such as Bacteroides fragilis and Enterobacteriaceae [61,62]. Recent evidence suggests this dysbiosis extends to the gut virome: the expansion of specific bacteriophages targeting SCFA-producing bacteria directly contributes to their decline. For instance, bacteriophage-mediated depletion of *F. prausnitzii* correlates with reduced butyrate production and enhanced gut inflammation, highlighting a trans-kingdom mechanism driving metabolic alterations [63,64].

The resulting deficit in SCFAs (butyrate, acetate, propionate) has multifaceted immunomodulatory consequences. SCFAs normally promote tolerance by inhibiting histone deacetylases (HDACs); butyrate, specifically, enhances histone acetylation at the promoters of *Foxp3* and *Il10*, thereby fostering regulatory T cell (Treg) differentiation and function [65]. Concurrently, SCFAs signal through G-protein coupled receptors (GPR43/GPR41) on immune cells, further supporting Treg and regulatory B cell (Breg) development [66]. In T1D, SCFA deficiency compromises these regulatory pathways, skewing the immune system toward a pro-inflammatory state. Furthermore, dysbiosis reduces secondary bile acids, impairing farnesoid X receptor (FXR) activation in dendritic cells, which subsequently enhances IL-12 production and Th1 differentiation [60,67,68].

### 4.2. Breakdown of the Intestinal Barrier and Resultant Immune Dysregulation

The dysfunction of the gut microbiome and the resulting metabolite alterations culminate in the breakdown of the intestinal barrier, a critical event that initiates a cascade of immune dysregulation leading to the loss of self-tolerance. This section delineates the pathway from barrier defect to autoimmune activation.

#### 4.2.1. Barrier Dysfunction and Antigen Translocation

The integrity of the intestinal barrier, maintained by epithelial cells, tight junction proteins (e.g., occludin, claudins), and a protective mucus layer, is compromised in T1D. Dysbiosis is a primary driver of this “leaky gut.” Reduced levels of short-chain fatty acids (SCFAs), particularly butyrate, diminish the stimulation of goblet cells, leading to impaired secretion of mucin 2 (MUC2) and a thinning of the inner mucus layer [66,69]. Concurrently, dysbiotic microbes and environmental triggers upregulate the expression of zonulin. Zonulin binds to the epidermal growth factor receptor (EGFR), activating the RhoA-Rock pathway, which induces actin cytoskeleton rearrangement and reduces occludin phosphorylation, ultimately dissociating tight junction proteins (Figure 4) [69,70,71].

Crucially, this barrier defect is often compounded by the enteric viral infections discussed in Section 3. These pathogens act synergistically with dysbiosis to further facilitate the translocation of microbial antigens, including bacterial peptides and lipopolysaccharide (LPS), as well as dietary proteins, from the gut lumen into the systemic circulation and mesenteric lymph nodes [46,47,48,49,50,51,72,73].

#### 4.2.2. Microbiota-Driven Immune Activation and Loss of Tolerance

The translocation of luminal antigens into the host system creates a pro-autoimmune environment by dysregulating both innate and adaptive immunity. The ensuing loss of tolerance occurs through multiple, non-mutually exclusive mechanisms.

First, the innate immune system is activated. Translocated LPS binds to Toll-like receptor 4 (TLR4) on dendritic cells (DCs) and macrophages, triggering the NF-κB pathway and the secretion of pro-inflammatory cytokines such as TNF-α, IL-6, and IL-1β [50,65]. These cytokines create a local and systemic inflammatory milieu that lowers the activation threshold for immune cells. Furthermore, gut-derived bacterial DNA can activate TLR9 on DCs, prompting the production of type I interferons (IFN-α/β), which can enhance MHC class I expression on β-cells, making them more susceptible to CD8+ T cell-mediated cytotoxicity [74].

Second, the dialogue between microbial signals and the immune system disrupts the delicate balance between effector and regulatory responses. The inflammatory environment, characterized by the cytokines mentioned above, promotes the polarization of naïve T cells towards pro-inflammatory T-helper 1 (Th1) and Th17 phenotypes, while simultaneously impairing the function and development of regulatory T cells (Tregs) [75,76]. The depletion of SCFAs in T1D further exacerbates this imbalance, as SCFAs are critical for the induction and suppressive function of Tregs via inhibition of histone deacetylases and G-protein coupled receptor signaling [65]. This leads to a state of uncontrolled effector T cell responses (Figure 4).

Third, loss of tolerance is driven by antigen-specific mechanisms such as molecular mimicry. Certain gut commensals, notably Parabacteroides distasonis, express peptides with structural homology to pancreatic β-cell antigens (e.g., the insulin B-chain epitope). This bacterial mimicry parallels the viral mechanisms described earlier, collectively expanding the pool of cross-reactive antigens that can activate naïve T cells in the inflammatory environment [42,68,77].

Finally, microbial metabolites exert epigenetic influences that can stabilize pathogenic immune cell phenotypes. SCFAs, for instance, enhance DNA methyltransferase (DNMT) activity, promoting methylation of pro-inflammatory gene promoters. Their reduction in T1D can lead to hypomethylation and increased expression of genes like IL-17 and TNF-α, thereby cementing a pro-inflammatory state and contributing to the long-term breakdown of immune tolerance [78,79,80].

## 5. Oxidative Stress, Mitochondrial Dysfunction, and Inflammatory Mechanisms

### 5.1. Oxidative Stress and Mitochondrial Dysfunction

Environmental factors contribute to T1D in genetically susceptible individuals by disrupting cellular homeostasis, with oxidative stress and mitochondrial dysfunction serving as central mechanistic links. Viral infections, in particular, induce a sustained interferon-alpha (IFN-α) response in pancreatic β-cells, triggering the production of mitochondrial-derived ROS (mtROS). A subset of β-cells acutely produces mtROS upon IFN-α exposure—a characteristic associated with healthier donor islets and potentially representing an efficient antiviral response that inadvertently contributes to immunogenicity [81]. This IFN-α-driven stress also impairs insulin production by inducing ER stress and disrupting mitochondrial function [39].

Environmental factors can also lead to the formation of advanced glycation end products (AGEs), which are elevated in children who progress to T1D. AGEs directly impair β-cell function by inducing mitochondrial abnormalities, including excess superoxide generation, a decline in ATP content, and loss of manganese superoxide dismutase (MnSOD) activity—leading to secretory defects and apoptosis [82,83].

β-cells are inherently vulnerable to oxidative stress due to their relatively low expression of antioxidant enzymes [81]. Mitochondrial dysfunction, characterized by defective oxidative phosphorylation and impaired ATP production, exacerbates this vulnerability [84]. This dysfunction is an early event, as integrative multi-omics analyses of children at risk for T1D revealed abnormalities in lipid metabolism and increased intracellular ROS accumulation prior to seroconversion [85]. The sensitivity of β-cells to mtROS is associated with genetic T1D risk loci, suggesting that intrinsic mitochondrial defects can predispose β-cells to immune-mediated destruction [81,84].

Crucially, the impact of ROS extends to the immune compartment, playing a dual role. In infiltrating macrophages, NADPH oxidase (NOX)-derived superoxide creates a pro-inflammatory environment; its absence skews macrophages toward a protective M2 phenotype [86,87]. Conversely, in T cells, ROS signaling acts as a critical checkpoint for differentiation. Loss of T cell-derived superoxide impairs regulatory T cell (Treg) function and exacerbates diabetogenic effector responses, highlighting the complex, cell-specific nature of redox regulation in T1D [88,89].

### 5.2. Inflammatory Mechanisms

As detailed in Section 4, gut barrier disruption allows the translocation of microbial endotoxins (e.g., LPS), triggering chronic low-grade inflammation. This is clinically evident in at-risk individuals, who exhibit pro-inflammatory metabolic profiles and heightened innate immune activation well before autoantibody appearance [90,91,92].

The dialogue between environmental triggers and the innate immune system is fundamental. Viral infections (particularly enteroviruses) lead to local inflammation in the pancreas or visceral adipose tissue (VAT), characterized by the release of proinflammatory cytokines (e.g., type I interferons) and chemokines [93,94]. This inflammation promotes β-cell stress, enhances their immunogenicity, and facilitates recruitment and activation of autoreactive T cells [93,95]. Macrophages are among the first innate immune cells to infiltrate the islets, releasing ROS and cytokines (IL-1β, TNF-α) that create a destructive inflammatory environment [87].

Chemokines amplify this process by directing immune cell trafficking to the islets. β-cells themselves express chemokines such as CXCL10 upon stress or cytokine exposure, initiating a vicious cycle of mutual attraction between immune cells and antigen-presenting cells within the islet microenvironment [95,96]. The resulting insulitis—a hallmark of T1D—is characterized by progressive inflammatory infiltration leading to β-cell destruction [2,42].

Notably, gut dysbiosis actively fuels this inflammatory fire. Beyond the barrier defects previously discussed, dysbiosis in T1D is directly correlated with intestinal inflammation markers (e.g., human beta-defensin-2 [HBD2], calprotectin). This altered microbial community shifts the mucosal immune balance, further propagating the systemic autoimmunity that targets the pancreas [79,97,98,99].

## 6. Epigenetic Modifications Mediated by Environmental Triggers

While genetic susceptibility provides the foundational risk for Type 1 Diabetes (T1D), the incomplete concordance in monozygotic twins highlights the critical role of environmental factors and their biological intermediaries—epigenetic modifications [80,100]. Epigenetics refers to heritable changes in gene expression that do not involve alterations to the underlying DNA sequence. These modifications, including DNA methylation, histone modifications, and non-coding RNA regulation, represent a dynamic mechanism through which environmental exposures can persistently influence immune function and β-cell biology, thereby contributing to T1D pathogenesis [80,101].

### 6.1. DNA Methylation: A Key Epigenetic Regulator in T1D

DNA methylation, the addition of a methyl group to cytosine bases in CpG dinucleotides, is the most extensively studied epigenetic mark in T1D. Genome-wide analyses have revealed distinct methylation profiles in peripheral blood cells of individuals with T1D compared to healthy controls [100,102,103]. These changes are not random but are often enriched at genes central to the disease process. For instance, differential methylation has been consistently identified in regions regulating immune tolerance (e.g., *FOXP3*, *CTLA4*), antigen presentation (e.g., *HLA* genes), and β-cell function (e.g., *INS*) [80,104].

Crucially, epigenetic changes can be triggered by environmental factors. Pro-inflammatory cytokines, such as interferon-alpha (IFN-α), have been shown to induce DNA demethylation at specific loci in human islets, leading to the upregulation of genes involved in immune pathways. This process can involve mechanisms like the *PNPT1/miR-26a/TET2* axis, demonstrating a direct link between an inflammatory environment and epigenetic remodeling in β-cells [105]. Intriguingly, in utero exposures leave lasting epigenetic imprints; for instance, maternal smoking is associated with specific offspring methylation scores that paradoxically correlate with a reduced risk of T1D, highlighting the complex, context-dependent nature of early-life programming [106].

The relationship between genetics and epigenetics is bidirectional. Single Nucleotide Polymorphisms (SNPs) associated with T1D risk can function as methylation Quantitative Trait Loci (mQTLs), influencing the methylation status of nearby CpG sites. This has been demonstrated for several susceptibility genes, including *INS* (rs689), *IL2RA*, and *PTPN22* [104,107,108]. For example, the T1D-risk genotype at the *INS* locus is associated with higher methylation levels in its promoter region, potentially affecting its expression and contributing to disease risk [105,108]. This interplay suggests that part of the genetic risk for T1D is mediated by epigenetic mechanisms, creating a permissive genomic environment for disease initiation upon environmental triggers.

### 6.2. Histone Modifications and Non-Coding RNAs

Histone modifications (e.g., H3K4me3, H3K27ac) govern chromatin accessibility. Dysregulation of these marks has been implicated in T1D complications, such as the sex-specific suppression of GPER in gastrointestinal tissues [109]. Additionally, enzymes like Peptidylarginine Deiminase (PADs) play a dual pathogenic role: they alter chromatin structure via histone citrullination and simultaneously generate immunogenic neo-epitopes, fueling the autoimmune fire [110].

Non-coding RNAs, particularly microRNAs (miRNAs), act as post-transcriptional epigenetic regulators. Altered circulating miRNA profiles have been identified in T1D patients and are proposed as potential biomarkers for disease progression and complications [109,111]. For instance, the circular RNA circ-0000953, which is downregulated in diabetic nephropathy (DN), regulates podocyte autophagy by sponging miR-665-3p, illustrating a complex regulatory network involving non-coding RNAs in diabetes complications [112].

### 6.3. Epigenetics in T1D Complications and Clinical Translation

Epigenetic modifications are also implicated in the development of T1D complications. Specific methylation patterns in blood have been associated with and may predict the onset of proliferative diabetic retinopathy (PDR) and diabetic kidney disease (DKD) [102,103,111]. Multi-omics studies integrating DNA methylation with genetic variants and circulating proteins have identified epigenetic markers that improve the prediction of kidney failure in T1D patients, offering avenues for risk stratification and early intervention [107,111].

The reversible nature of epigenetic marks makes them attractive therapeutic targets. Strategies aiming to modulate aberrant epigenetic patterns, such as using inhibitors of specific enzymes like PAD or employing antisense oligonucleotides (e.g., GapmeRs) to target pathogenic long non-coding RNAs, are being explored in preclinical models and hold promise for future clinical applications [110,113].

In summary, epigenetic mechanisms provide a critical molecular link between genetic predisposition and environmental factors in T1D. DNA methylation, histone modifications, and non-coding RNAs work in concert to modulate the expression of genes governing immune regulation and β-cell survival. These modifications can be altered by viral infections, diet, toxins, and other exposures, potentially explaining the rising incidence of T1D and its variable presentation. A deeper understanding of the epigenome not only elucidates disease etiology but also opens new frontiers for biomarker discovery and the development of novel, personalized therapeutic strategies to prevent or halt T1D and its devastating complications.

## 7. Gene–Environment Interactions in T1D Pathogenesis

The pathogenesis of Type 1 Diabetes (T1D) exemplifies a complex disease model where genetic susceptibility provides a baseline risk that is dynamically influenced by environmental factors [114,115]. While Genome-Wide Association Studies (GWAS) have identified over 60 susceptibility loci—with the HLA region conferring the highest risk—these variants do not fully account for the disease’s heritability. This gap, known as “missing heritability,” highlights the indispensable role of environmental exposures. T1D is thus best understood as a result of the interplay between “nature and nurture,” where environmental triggers precipitate autoimmune responses in genetically predisposed individuals [115,116].

### 7.1. Genetic Susceptibility: Setting the Stage

The strongest genetic associations for T1D reside within the Major Histocompatibility Complex (MHC), particularly *HLA* class II genes (e.g., *DRB1*, *DQA1*, *DQB1*), which are pivotal in antigen presentation to T-cells [95,116]. Non-*HLA* genes further modulate risk, often involving pathways central to immune regulation (e.g., *PTPN22*, *CTLA4*, *IL2RA*) and β-cell function and survival (e.g., *INS*, *CTSH*) [115,116,117]. However, the incomplete concordance of T1D in monozygotic twins (∼40–50%) provides compelling evidence that non-genetic factors are indispensable for disease manifestation [114,115]. This indicates that genetic risk alleles are not deterministic but instead increase vulnerability to environmental perturbations.

### 7.2. Environmental Triggers: The External Catalysts

A diverse array of environmental factors has been implicated in T1D pathogenesis. These include viral infections (e.g., enteroviruses), dietary components during early life (e.g., timing of gluten introduction, cow’s milk proteins), the gut microbiome, vitamin D levels, and exposure to pesticides or other chemicals [114,117,118,119,120]. Large prospective cohort studies, such as The Environmental Determinants of Diabetes in the Young (TEDDY), have been instrumental in evaluating these candidates, revealing that the pathways to islet autoimmunity and clinical T1D are likely multifactorial and heterogeneous [118,119,121]. For instance, the TEDDY study found that the timing of gluten introduction influences islet autoimmunity risk, with both very early and late introduction associated with altered risk compared to introduction between 4 and 9 months of age [120]. Furthermore, early-life diet can have intergenerational effects; exposure to the A1 variant of beta-casein in cow’s milk was shown to increase T1D incidence in non-obese diabetic (NOD) mice over several generations [117].

The gut microbiome represents a particularly critical interface between the host and environment. Altered gut microbial composition (dysbiosis), characterized by features such as a higher *Firmicutes*/*Bacteroidetes* ratio or reduced abundance of *Bifidobacterium*, has been associated with T1D development in both human studies and animal models [73,120,122]. The microbiome is thought to influence mucosal integrity, immune tolerance, and systemic immunity through mechanisms including molecular mimicry, modulation of the gut immune system, and changes in short-chain fatty acid production [73,116]. Intriguingly, *HLA* haplotypes associated with T1D risk may also correlate with specific gut microbiome profiles, suggesting a direct genetic influence on the microbial environment, which in turn modulates disease risk [114,122].

### 7.3. Mechanisms of Interaction: Bridging Genes and Environment

The interplay between genetic predisposition and environmental exposures can be mediated through several mechanistic pathways:

Modulation of Immune Responses to Pathogens: Certain T1D risk alleles may alter the host’s immune response to viral infections. For example, genetic variants in immune response genes can predispose individuals to a hyper-reactive state upon infection or, conversely, facilitate viral persistence, both of which may contribute to the loss of immune tolerance [119]. Enteroviruses, in particular, have been studied for their potential to trigger islet autoimmunity in genetically susceptible children [118,119].

Activation of Endogenous Retroviral Elements: Environmental viruses can activate dormant human endogenous retroviruses (HERVs) in the genome. The envelope protein of the HERV-W family (*HERV-W-Env*) has been detected in the pancreata of T1D patients and exhibits pro-inflammatory and directly toxic effects on β-cells. This provides a novel mechanism whereby an environmental infection can unleash a latent genetic element with pathogenic potential [60].

Epigenetic Modification of Risk Loci: As discussed in Section 6, epigenetics translates environmental signals into gene expression changes. A prime example of Gene–Environment interaction is the *CTSH* locus: pro-inflammatory cytokines can induce DNA hypermethylation at this T1D risk gene, reducing its expression and promoting β-cell apoptosis. Crucially, the magnitude of this effect depends on the underlying *CTSH* genotype [117,122].

Gene–Environment Correlation (GxE) in Disease Incidence: Indirect evidence for interaction comes from observations that the strength of *HLA* associations with T1D may vary over different time periods, suggesting that changing environmental exposures can modulate the penetrance of genetic risk alleles across populations and generations [123].

In summary, T1D is not a disease of genetics or environment alone but a consequence of their intricate and continuous dialogue. Genetic makeup determines an individual’s susceptibility threshold, while environmental factors act as triggers that can lower this threshold and initiate the disease process. Understanding these interactions is crucial for explaining the “missing heritability,” identifying individuals at highest risk, and developing targeted prevention strategies [95,115]. Future research, leveraging large-scale longitudinal cohorts and multi-omics integration, will be key to delineating specific gene–environment pathways. This knowledge paves the way for precision medicine approaches, where risk assessment and interventions can be tailored based on an individual’s unique genetic and environmental profile, ultimately aiming to delay or prevent the onset of T1D [95,118].

## 8. Conclusions and Future Perspectives

This review synthesizes compelling evidence that environmental factors are pivotal drivers in the pathogenesis and progression of Type 1 Diabetes (T1D), interacting with genetic susceptibility to trigger and modulate the autoimmune destruction of pancreatic β-cells [1,2,99]. The evidence delineates a complex web of mechanisms through which persistent organic pollutants (POPs), heavy metals, air pollutants, viral infections, and gut microbiome dysbiosis contribute to disease development. These mechanisms are not isolated but are highly interconnected, converging on critical pathways including oxidative stress, mitochondrial dysfunction, epigenetic reprogramming, innate immune activation, and loss of intestinal barrier integrity [29,39,70,81,100]. The findings underscore that T1D is a quintessential example of a disease arising from gene–environment interactions, where external exposures act upon a permissive genetic background to initiate and accelerate the autoimmune process [114,115].

The clinical translation of this knowledge is already underway, with promising strategies emerging. These include targeted interventions such as pollutant exposure reduction, microbiome modulation (e.g., fecal microbiota transplantation, specific probiotics), antioxidant therapy, and the exploration of epigenetic modifiers and viral vaccines. Furthermore, the implementation of population-level measures, such as public health policies to reduce environmental pollutants and nutritional guidelines for early life, holds significant potential for primary prevention.

An additional methodological consideration is that socioeconomic status (SES), dietary patterns, and geographical context shape both environmental exposures and type 1 diabetes (T1D) risk, and thus may systematically bias epidemiological findings [124,125]. Children from lower-SES households are more likely to live near industrial sites or high-traffic roads, experience overcrowded housing, have limited access to fresh foods, and receive delayed medical care. Ref. [124] These factors correlate with higher exposures to air pollutants, persistent organic pollutants, and toxic metals, as well as a greater burden of infections and later recognition of T1D, making it difficult to disentangle the independent effects of specific environmental triggers [10,125,126,127]. Dietary patterns further modify gut microbiota composition, intestinal permeability, and micronutrient status (e.g., iron, zinc, vitamin D), and can interact with pollutants and viral infections in ways that are rarely captured by crude “diet quality” indicators [128,129,130]. Likewise, geographical variation in climate, urbanization, agricultural practices, and background infection pressure leads to spatial clustering of both exposures (e.g., PM_2.5_, pesticides, viral serotypes) and T1D incidence [131,132]. Studies relying on registry data, ecological exposure metrics, or single-time-point biospecimens may therefore be prone to unmeasured or residual confounding, collider bias (for example, when restricting to children who survive severe infections), and differential misclassification of exposure across SES strata or regions. Future epidemiological work should explicitly model SES, diet, and geography as both confounders and potential effect modifiers, incorporating individual- and neighborhood-level indicators and, where possible, applying modern causal-inference approaches to more rigorously estimate the causal contribution of specific environmental exposures to T1D.

However, several critical challenges and controversies remain, which define the agenda for future research:From Association to Causation: While epidemiological studies robustly link environmental factors to T1D, establishing definitive causal relationships in humans is complex. Future research must move beyond correlation by leveraging longitudinal cohorts from birth, like the TEDDY study, refs. [6,118] with more frequent and precise exposure assessments. Integrating multi-omics approaches (genomics, epigenomics, metabolomics, metagenomics) from serial samples will be essential to delineate the temporal sequence of events from exposure to immune activation and β-cell damage [85,108].The Exposome and Multi-Factorial Interactions: Individuals are exposed to a mixture of environmental factors simultaneously, not in isolation. A major future direction is to characterize the “T1D exposome”—the totality of exposures throughout life—and understand how these factors interact synergistically or antagonistically [9,28]. Advanced statistical models and machine learning algorithms will be necessary to decipher these complex interactions and identify critical windows of exposure, particularly during gestation, early childhood, and puberty [26,27].Resolving Mechanistic Complexity: The precise molecular mechanisms linking specific exposures to the breakdown of immune tolerance require further elucidation. Key unanswered questions include the exact role of epigenetic modifications as a persistent memory of environmental insults and the relative contribution of direct β-cell toxicity versus immune dysregulation. Research should prioritize human-relevant models, such as humanized mice, stem-cell-derived islets, and sophisticated in vitro systems, to validate mechanisms identified in animal models [6,33,40,50,100].Towards Personalized Prevention and Therapy: The variable individual response to environmental triggers, influenced by genetics, epigenetics, and microbiome composition, calls for a personalized medicine approach. Future efforts should focus on developing integrated risk scores that combine genetic, epigenetic, and environmental data to identify individuals at the highest risk. This will enable targeted, cost-effective prevention trials. Moreover, the therapeutic potential of targeting environmental mechanisms, such as using short-chain fatty acid derivatives or engineered probiotics to restore immune tolerance, represents a promising frontier for intervention in pre-symptomatic or new-onset T1D [65,95,133].Interdisciplinary Collaboration: Addressing the multifaceted challenge of environment–T1D interactions demands unprecedented collaboration across disciplines, including epidemiology, immunology, toxicology, microbiology, bioinformatics, and public policy. Only through such integrated efforts can we translate mechanistic insights into tangible strategies to mitigate the global burden of T1D [107].

Future Perspectives: To further advance the field, a focused concluding direction is warranted to address existing knowledge gaps. First, large-scale, long-term longitudinal cohort studies with standardized exposure assessment and repeated biological sampling are urgently needed to establish causal links between specific environmental triggers and T1D onset/progression. Second, integrating exposomics (comprehensive characterization of all lifelong environmental exposures) with systems biology approaches will enable the dissection of complex gene–environment–epigenome interactions, uncovering novel pathogenic pathways. Third, there is a critical need to validate mechanistic findings from animal and in vitro models in human populations, particularly focusing on key intermediates like oxidative stress markers, cytokine profiles, and gut barrier integrity. These research directions will not only refine our understanding of T1D etiology but also accelerate the development of targeted prevention strategies and personalized interventions for at-risk populations.

In conclusion, genetic predisposition lays the foundation for T1D, but environmental factors are the key drivers. A better understanding of these environmental factors and the molecular pathways they affect will not only provide a clearer picture of T1D etiology but also offer a roadmap for prevention and intervention. The future of T1D research lies in embracing this complexity to develop precise, effective, and sustainable strategies that protect susceptible individuals from the environmental triggers of our modern world.

## Figures and Tables

**Figure 1 ijms-26-11613-f001:**
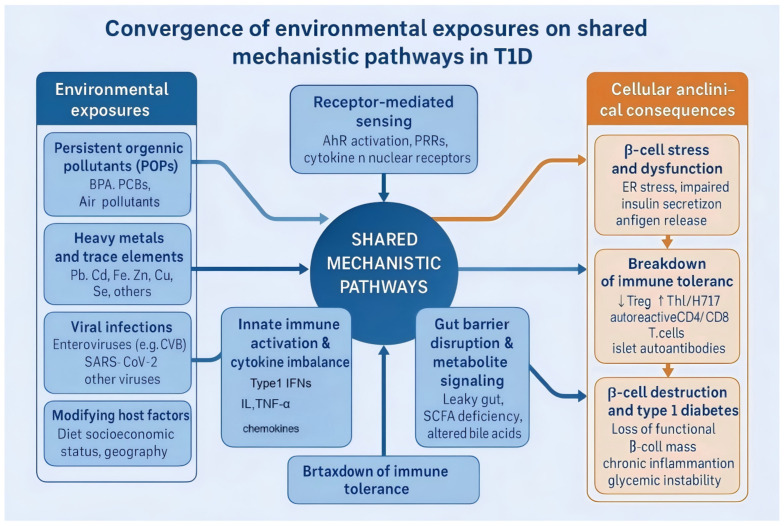
Major environmental exposures—including persistent organic pollutants (POPs), heavy metals and trace elements, air pollutants, viral infections, and alterations in the gut microenvironment, together with modifying host factors—converge on shared mechanistic pathways such as receptor-mediated sensing (e.g., AhR activation), oxidative stress and mitochondrial dysfunction, epigenetic reprogramming, innate immune activation with cytokine imbalance, and disruption of the gut barrier and metabolite signaling. These processes drive β-cell stress and dysfunction and breakdown of immune tolerance, ultimately leading to progressive β-cell destruction and clinical type 1 diabetes.

**Figure 2 ijms-26-11613-f002:**
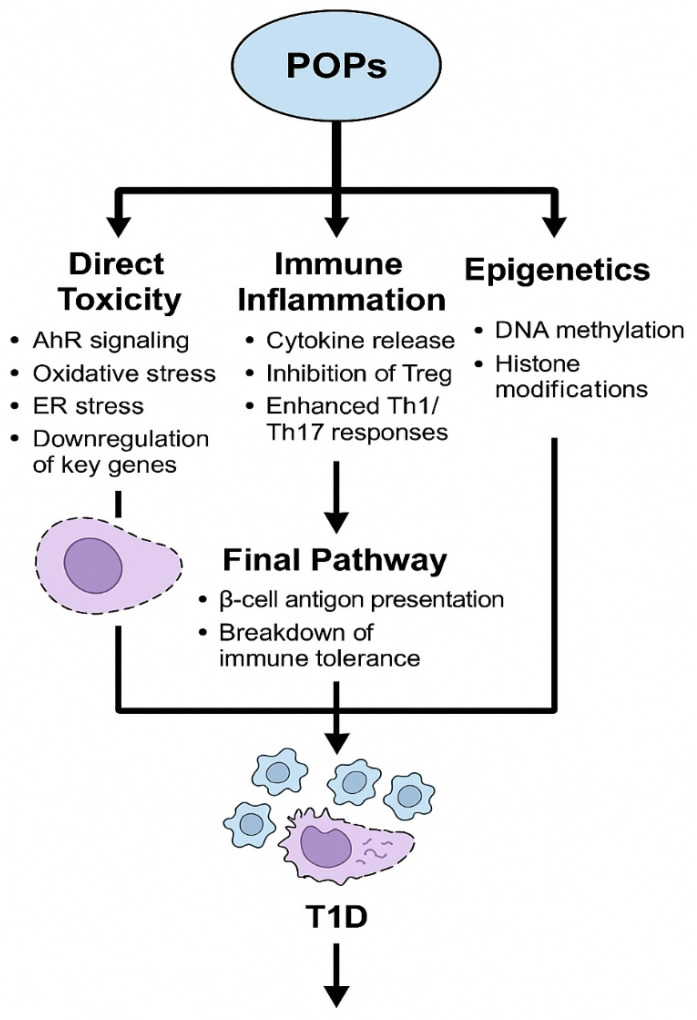
Schematic diagram illustrating the proposed mechanistic pathways by which persistent organic pollutants (POPs) contribute to the pathogenesis of type 1 diabetes (T1D).

**Figure 3 ijms-26-11613-f003:**
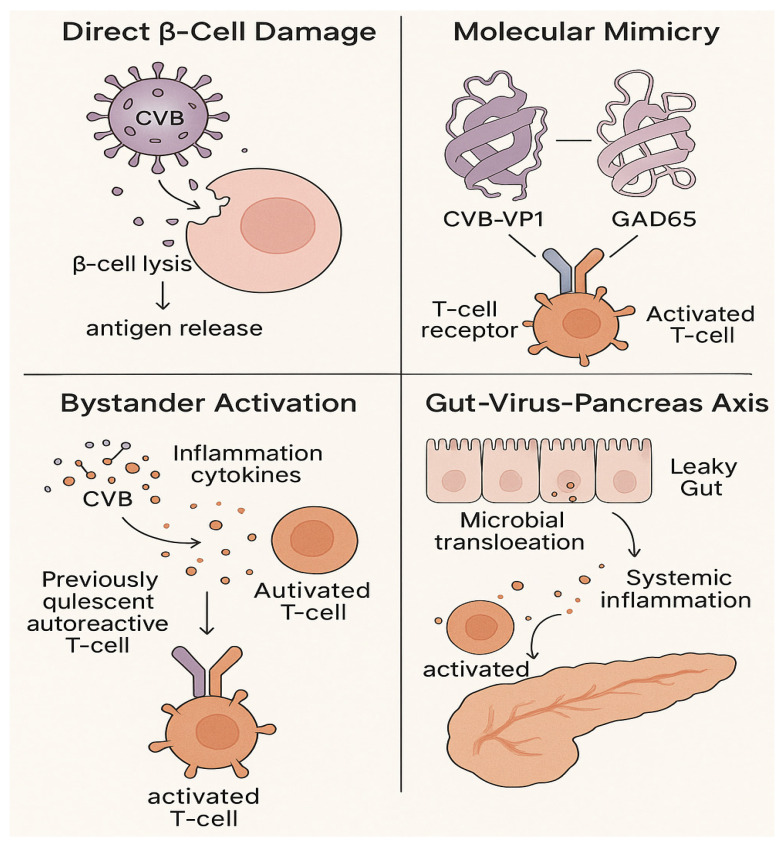
Mechanisms of viral-induced type 1 diabetes. Viruses contribute to disease pathogenesis through (1) direct β-cell damage and antigen release, (2) molecular mimicry leading to cross-reactive T cells, (3) bystander activation in a pro-inflammatory milieu, and (4) disruption of the gut-pancreas axis.

**Figure 4 ijms-26-11613-f004:**
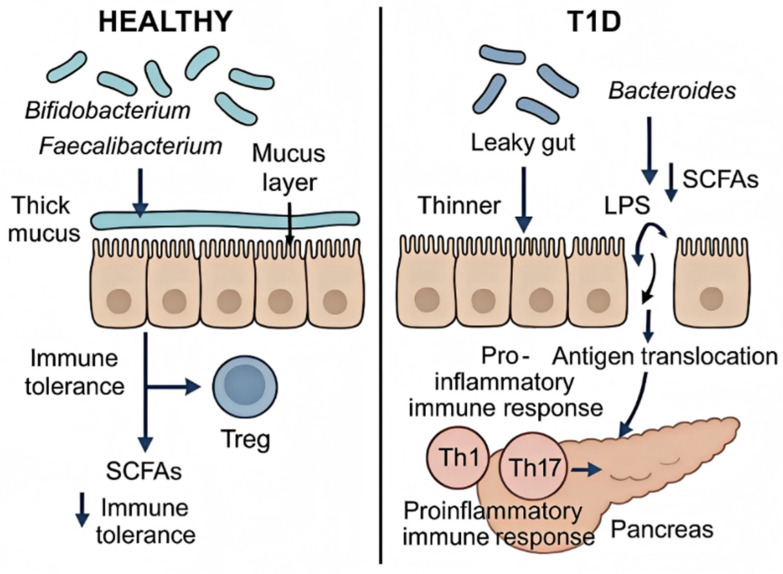
The role of gut microenvironment dysbiosis in T1D pathogenesis. A shift from a healthy to a dysbiotic gut microbiome, loss of barrier integrity, and altered metabolite production contribute to immune dysregulation and the breakdown of tolerance to pancreatic β-cells.

**Table 1 ijms-26-11613-t001:** Summary of Key Environmental Pollutants and Their Proposed Mechanisms in Type 1 Diabetes (T1D).

Pollutant Class	Representative Agent(s)	Proposed Primary Mechanism(s)	Association with T1D
Persistent Organic Pollutants (POPs)	Bisphenol A (BPA)	Endocrine disruption; Immune dysregulation; Metabolic alteration; AhR pathway activation inducing oxidative stress & inflammation	Increased risk
	Polychlorinated Biphenyls (PCBs), Organochlorine Pesticides (e.g., p,p′-DDE)	Direct β-cell toxicity (impaired insulin secretion); AhR pathway activation; Induction of oxidative stress & β-cell apoptosis	Increased risk
Heavy Metals	Lead (Pb)	Mitochondrial dysfunction; Disruption of insulin signaling; Promotion of oxidative stress	Increased risk (associated with poorer glycemic control)
	Cadmium (Cd)	Tissue accumulation; Direct pancreatic β-cell damage	Increased risk
Air Particulate Matter	PM_2.5_/PM_10_	Systemic inflammation; Oxidative stress; Impaired insulin sensitivity; Activation of pro-inflammatory pathways	Increased risk (associated with elevated HbA1c & hypoglycemia)
	Ozone (O_3_)	Prenatal immune disruption; Oxidative stress; Impaired fetal immune cell differentiation and β-cell development; Indirect effects via maternal metabolic alterations (e.g., gestational diabetes)	Increased risk (maternal exposure associated with higher T1D risk in offspring)
Trace Elements	Iron (Fe)	Oxidative stress (U-shaped response); Gene–environment interactions	Complex (U-shaped association)
	Zinc (Zn)	Immune system integrity; Insulin metabolism; Antioxidant defense	Deficiency linked to increased risk
	Copper (Cu)	Disruption of metabolic homeostasis; Promotion of oxidative stress (Fenton-like reactions)	Dysregulation observed (elevated in new-onset T1D)

## Data Availability

No new data were created or analyzed in this study. Data sharing is not applicable to this article.

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
