# Peer review of "Environmental Mechanisms Influencing the Pathogenesis and Progression of Type 1 Diabetes"

_ijms, 2025, doi:10.3390/ijms262311613_

Round 1
Reviewer 1 Report
Comments and Suggestions for Authors
This manuscript presents a comprehensive and well-organized systematic review addressing the environmental mechanisms contributing to the pathogenesis and progression of Type 1 diabetes (T1D). The topic is highly relevant, timely, and of significant scientific and public health interest, given the growing global burden of autoimmune diseases and the increasing exposure to environmental toxicants. The paper is well-structured, the writing is generally clear, and the abstract provides a concise overview of the study’s objectives and findings.
The inclusion of molecular and epidemiological evidence enhances the depth of the review. However, several issues require clarification and refinement before the manuscript can be considered for publication. These include refining mechanistic discussions, standardizing referencing style, and enhancing figure/table presentation to meet the journal’s quality standards.
Abstract: Add a sentence summarizing the principal conclusion or implication (e.g., “Identifying these mechanistic pathways highlights potential targets for preventive interventions in at-risk populations.”).
Section 2.1: The transitions between experimental, epidemiological, and mechanistic findings could be smoother. Consider subheadings such as 2.1.1 Human Studies, 2.1.2 Experimental Studies, and 2.1.3 Mechanistic Insights for better readability. When discussing dose-dependent BPA effects, provide clear concentration units and specify exposure duration for reproducibility.
Abbreviations: Provide a consolidated list of abbreviations or define each abbreviation at first mention (e.g., GMB, AhR, NOD).
Methodological Transparency (PRISMA Compliance): The manuscript states that it follows PRISMA guidelines, but the PRISMA flow diagram and study selection criteria (inclusion/exclusion, number of records screened, reasons for exclusion, etc.) are not presented. Include a PRISMA flowchart and a detailed table summarizing key characteristics of the included studies (e.g., study type, sample size, exposure type, outcomes assessed, quality score).
Mechanistic Integration: While the discussion of POPs (particularly BPA and PCBs) is comprehensive, the manuscript would benefit from greater integration across mechanistic pathways (e.g., linking oxidative stress, epigenetic alterations, and immune dysregulation into a unified conceptual model). Add a schematic figure summarizing how various environmental exposures converge on common molecular targets (AhR activation, mitochondrial ROS generation, cytokine imbalance, etc.).
Scope and Balance of Coverage: The current version focuses extensively on POPs, while other environmental factors mentioned in the abstract (e.g., heavy metals, air pollutants, viral infections, and gut microbiota alterations) are not yet elaborated in equal depth. Ensure that subsequent sections maintain a balanced discussion of each environmental category, or clarify that the current section represents only one component of a multipart review.
Critical Appraisal of Included Studies: Although quality assessment tools (NOS, ARRIVE, AMSTAR 2) are mentioned, the results of these evaluations are not summarized. Please, include a concise paragraph or supplementary table showing the methodological quality of included studies and potential sources of bias.
Clarification of Epidemiological Evidence: The section cites large-scale studies (e.g., >44,000 pediatric/adolescent T1D patients) and quantitative associations (e.g., PM quartiles, ORs, HbA1c differences), but lacks contextual discussion of confounders (e.g., socioeconomic factors, urban density, co-exposures) or biological plausibility. Add 1–2 sentences clarifying whether these associations persisted after adjustment for potential confounding variables and highlight limitations of such population-based analyses.
Incomplete Mechanistic Description in Section 2.3.3: The paragraph on “Impairment of Pancreatic Islet Function” ends abruptly (“PMâ‚‚.â‚… exposure in T1D mice increased lung lipid…”). This suggests missing continuation or incomplete description of the findings. Complete this subsection by elaborating on the downstream effects—e.g., β-cell oxidative damage, mitochondrial impairment, apoptosis, or inflammatory infiltration of pancreatic tissue.
In the section: 2.3.4. Pollutant-Specific Effects: Expand the O₃ and ASD sections by including mechanistic explanations (e.g., oxidative stress, cytokine signaling, placental inflammation, or immune tolerance disruption) to achieve balance and continuity with preceding subsections (2.3.1–2.3.3). Provide interpretative context—explain that even small relative risks can translate into meaningful public health impacts given widespread exposure and cumulative lifetime risk. Clarify this apparent paradox. Discuss whether the delay may result from immune modulation (e.g., transient activation of IFN-γ or compensatory Treg regulation) or from experimental factors (dose, exposure route, or mouse strain). Explicitly acknowledge this as an exception or a model-specific finding.
Language and Style: The writing is technically sound but could be improved for conciseness and readability. Some sentences are overly complex, and grammatical issues appear. The English could be improved to more clearly express the research.
Future Perspectives: Consider adding a short concluding paragraph outlining knowledge gaps and future research directions (e.g., need for longitudinal cohort studies, integration of exposomics, and systems biology).
The manuscript presents valuable insights and a strong scientific foundation, but it requires moderate restructuring and additional methodological clarity to reach publishable quality. Addressing the above comments will significantly enhance the rigor, coherence, and impact of the review.
Comments on the Quality of English Language
The English could be improved to more clearly express the research.
Reviewer 2 Report
Comments and Suggestions for Authors
This manuscript presents a comprehensive and well-organized review exploring how environmental factors contribute to the onset and progression of T1D. The paper synthesizes data from epidemiological, clinical, and experimental studies and provides mechanistic insights into oxidative stress, mitochondrial dysfunction, immune dysregulation, and epigenetic reprogramming. Overall, the review is timely, informative, and methodologically sound, making a valuable contribution to understanding the multifactorial etiology of T1D. However, several aspects could be strengthened to improve critical depth, conciseness, and interpretative balance.
The authors compiled findings from multiple scientific domains, including toxicology, virology, genetics, and microbiome research. The paper bridges mechanistic and clinical data, providing an understanding of environmental contributions to T1D.
Figures and tables clearly illustrate complex mechanisms, such as pollutant-induced β-cell toxicity and viral–immune interactions. The conceptual model linking oxidative stress, immune imbalance, and gut barrier dysfunction is particularly well-articulated.
The discussion highlights the implications for preventive strategies, such as exposure reduction, microbiome modulation, and personalized medicine approaches. The inclusion of recent viral findings (e.g., SARS-CoV-2 and T1D onset) adds current and practical relevance.
The manuscript tends to emphasize supportive studies while offering limited discussion of contradictory or null findings. A more balanced critique of evidence strength and study limitations would improve objectivity.
However, the manuscript can be improved:
- It is stated in the abstract that this work is a systematic review, but, in fact, it is not!! This work is a narrative review.
- No methodology is present in the manuscript describing how the search was performed.
- The paper does not include quantitative analyses. The authors could summarize effect sizes or confidence intervals where available to strengthen the evidence base.
- Covering pollutants, metals, air quality, viral infections, microbiota, and epigenetics within one paper results in uneven depth across sections. In this way, the discussion could benefit from condensing certain mechanistic details and focusing more deeply on key environmental categories.
- The influence of socioeconomic status, diet, or geographical variation on environmental exposures and T1D risk is underexplored. A brief commentary on how these factors could bias epidemiological findings would enhance interpretive rigor.
- Some sections are overly dense, with long paragraphs and repetitive mechanistic explanations.
- Minor grammatical inconsistencies and inconsistent figure referencing should be corrected before publication, such as:
the sentence in the Abstract - “ hile genetic predisposition establishes disease susceptibility, environmental factors play a crucial role in triggering and modulating the autoimmune process.” – First word should be “While”.
- Ensure consistent citation formatting and numbering alignment throughout.
- Consider condensing Figures 2–4 or merging overlapping mechanistic pathways for visual clarity.
- Where possible, include quantitative evidence summaries (e.g., odds ratios, relative risks) to substantiate associations.
Round 2
Reviewer 2 Report
Comments and Suggestions for Authors
The authors have made a deep revision to the manuscript according with all recomendations.